# Leopard: A Vision Language Model for Text-Rich Multi-Image Tasks

**Mengzhao Jia[1][*][†]  Wenhao Yu[2][†]  Kaixin Ma[2][†]  Tianqing Fang[2]  Zhihan Zhang[1][*]  Siru Ouyang[3][*]**
**Hongming Zhang[2]  Dong Yu[2]  Meng Jiang[1]**

**[1]University of Notre Dame   [2]Tencent AI Seattle Lab   [3]UIUC**

**[1]mjia2@nd.edu;   [2]wenhaowyu@global.tencent.com**

**[*] Interns at Tencent AI Seattle Lab, [†] Core Contributors**

**Reviewed on OpenReview:** `https://openreview.net/forum?id=R2rasAEPVi`

## Abstract

Text-rich images, where text serves as the central visual element guiding the overall understanding, are prevalent in real-world applications, such as presentation slides, scanned documents, and webpage snapshots. Tasks involving multiple text-rich images are especially challenging, as they require not only understanding the content of individual images but reasoning about inter-relationships and logical flows across multiple visual inputs. Despite the importance of these scenarios, current multimodal large language models (MLLMs) struggle to handle such tasks due to two key challenges: (1) the scarcity of high-quality instruction tuning datasets for text-rich multi-image scenarios, and (2) the difficulty in balancing image resolution with visual feature sequence length. To address these challenges, we propose LEOPARD, an MLLM tailored for handling vision-language tasks involving multiple text-rich images. First, we curated about one million high-quality multimodal instruction-tuning data, tailored to text-rich, multi-image scenarios. Second, we proposed an adaptive high-resolution multi-image encoding module to dynamically optimize the allocation of visual sequence length based on the original aspect ratios and resolutions of images.

Experiments on a diverse set of benchmarks reveal that our model consistently outperforms state-of-the-art systems, such as Llama-3.2 and Qwen2-VL, in challenging text-rich, multi-image evaluations. Remarkably, our approach achieves outstanding performance using only 1.2M training instances, all of which are fully open-sourced, demonstrating both high efficiency and effectiveness compared to models trained on large-scale in-house data. Our code and data are available at `https://github.com/tencent-ailab/Leopard`.

## 1 Introduction

Multimodal large language models (MLLMs) have revolutionized vision-language tasks, driving advancements in various areas such as image captioning and object detection (Zhang et al., 2024; Zang et al., 2024). These improvements extend to applications involving *text-rich images* where text serves as the primary visual element guiding image comprehension, such as visual document understanding (Mathew et al., 2021) and scene text recognition (Singh et al., 2019b). Traditional OCR-based pipelines in these text-rich visual scenarios are being replaced by end-to-end approaches that directly encode intertwined multimodal inputs (Wu et al., 2023b; Zhang et al., 2023; Tang et al., 2024), leading to improved accuracy in handling text-rich images.

Despite these advancements, the majority of existing open-source MLLMs, like LLaVAR (Zhang et al., 2023) and mPlug-DocOwl-1.5 (Hu et al., 2024a), have primarily focused on optimizing performance for text-rich *single-image tasks*. This focus inherently limits their applicability in many real-world scenarios, where tasks often involve *multiple inter-connected images*. For instance, multi-page visual document understanding

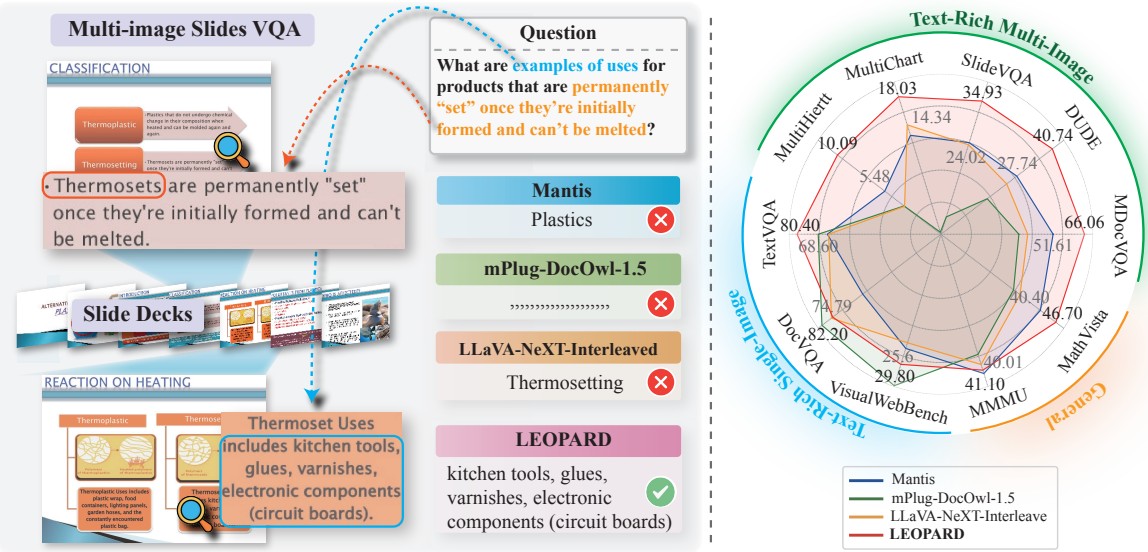

Figure 1: Left: A demonstration of a text-rich multi-image task, where models must reason across multiple images to answer correctly. LEOPARD generates the correct answer, while baselines fail. Right: LEOPARD outperforms three baselines on text-rich multi-image benchmarks by a large margin, while maintaining comparable performance on single-image and general evaluations.

requires integrating information spread across different pages to capture the logical flow across the whole document (Tito et al., 2022; Landeghem et al., 2023). To understand presentation slides, grasping the overarching narrative requires understanding multiple slides with unique but interrelated content (Tanaka et al., 2023). These vision-language tasks on multiple text-rich images require advanced capabilities that go beyond merely recognizing text and visuals within a single image; they involve understanding and reasoning about relationships and logical flows across multiple visual inputs. While some models – such as OpenFlamingo (Awadalla et al., 2023), VILA (Lin et al., 2023), Idefics2 (Laurençon et al., 2024b) – have made strides toward supporting multi-image inputs, they mainly focus on scenarios with natural images but fall short in understanding sequences of *text-rich images* with interrelated textual and visual information. We plot the performance of representatives of the aforementioned models in Figure 1. Upon examining their training data and model architecture, we identified two primary limitations.

First, there is a scarcity of high-quality instruction tuning datasets on text-rich multi-image scenarios. Existing visual instruction tuning datasets for text-rich images are predominantly based on single-image inputs (Kafle et al., 2018; Singh et al., 2019b; Masry et al., 2022; Tang et al., 2024), which limits the model ability to generalize and reason across multiple images. Second, in text-rich multi-image scenarios, there is a challenge of balancing image resolution and sequence length limitations. Many general-domain MLLMs adopt the low-resolution settings of pre-trained visual encoders (Lin et al., 2023; Jiang et al., 2024). However, for text-rich images, such as scientific reports, recognizing text content becomes difficult at low resolutions. While some approaches overcome this in single-image settings by splitting the original image to preserve high-resolution details (Liu et al., 2024a; Hu et al., 2024a), this approach is less effective when applied to multiple images, as it quickly exceeds model's maximum sequence length. Moreover, compressing such long-sequence representations into shorter ones leads to significant information loss, thereby degrading model performance (Awadalla et al., 2023; Laurençon et al., 2023). Thus, a critical balance must be struck between maintaining sufficient visual detail and keeping sequence lengths manageable. In this paper, we introduce a novel multimodal large language model, **Leopard**, designed for complex *text-rich, multi-image* tasks. To train LEOPARD, we curated approximately one million high-quality multimodal instruction-tuning samples, tailored for text-rich, multi-image contexts. This dataset was developed through a combination of new multi-image data collection from the web with augmentation of question-answer pairs, transformation of unimodal text data into multi-image formats, and reformatting single-image resources by employing an assembly strategy. Ultimately, the dataset encompasses three essential domains prevalent in real-world scenarios: (1) multi-page

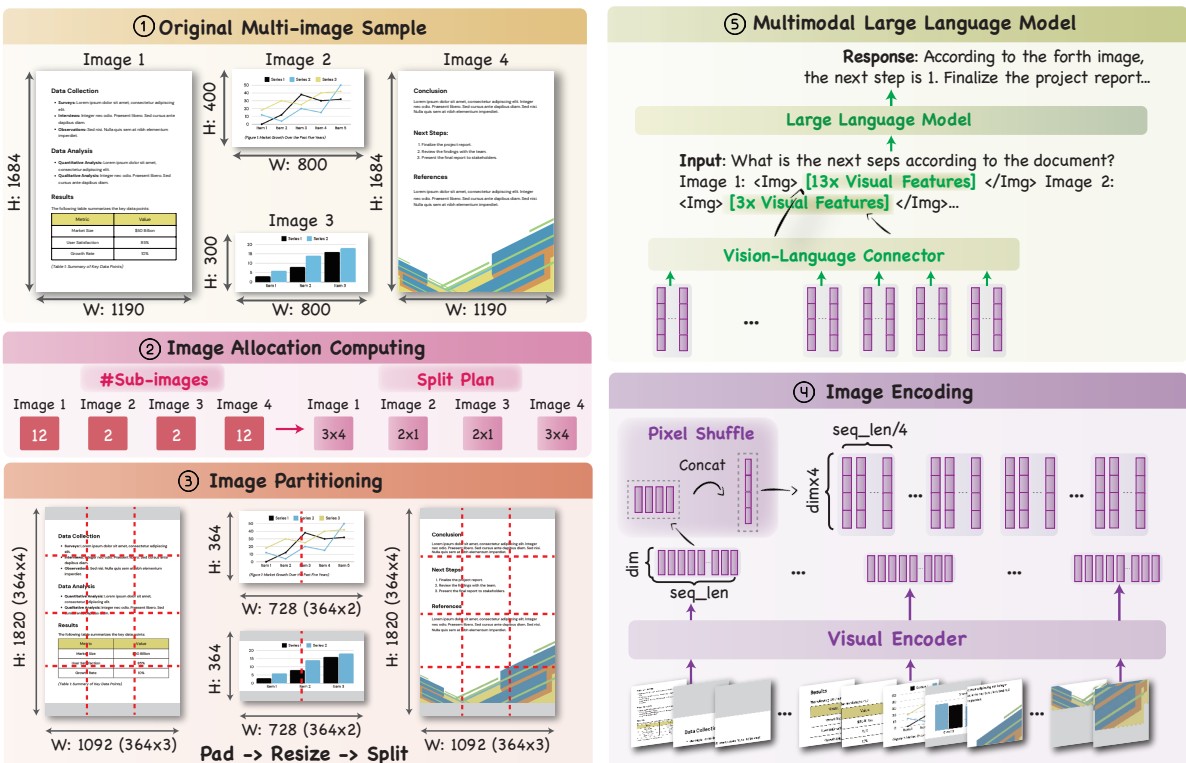

Figure 2: The overall model pipeline. Given ① raw image inputs, ② we first compute the optimal allocation of sub-image numbers and splitting strategy for all images based on their resolution and aspect ratio. ③ The images undergo padding, resizing, and splitting operations. ④ Both sub-images and resized original images are then encoded into a sequence of visual features. These sequences subsequently undergo a pixel shuffle operation that concatenates every four features. ⑤ The visual features are projected into the language embedding space via a vision-language connector. Finally, the large language model then integrates these visual and language embeddings to generate responses.

documents, (2) multi-charts and multi-tables, and (3) webpage sequences, reflecting the complex, multimodal demands of modern digital information. In addition, to enable high-resolution encoding in multi-image inputs, we equipped LEOPARD with an **adaptive high-resolution multi-image encoding module**. Specifically, it dynamically optimizes the allocation of visual sequence length based on the original aspect ratios and resolutions of the input images. We then apply pixel shuffling to losslessly compress (Chen et al., 2024) long visual feature sequences into shorter ones. This approach allows the model to accommodate multiple high-resolution images without compromising detail or clarity.

We conducted experiments on 12 vision-language benchmark datasets, evaluating LEOPARD from multiple perspectives. Consistent improvements were observed when training LEOPARD with two distinct base model architectures: LLaVA and Idefics2. Our results demonstrate LEOPARD's superior performance on 5 text-rich, multi-image benchmarks, outperforming the best open-source MLLM by an average of **+8.5** points. Moreover, LEOPARD remains highly competitive in text-rich single-image tasks and general-domain vision-language benchmarks, achieving comparable results to state-of-the-art MLLMs without extensive fine-tuning. Further ablation studies confirm the effectiveness of our instruction-tuning dataset and the adaptive high-resolution encoding module. These findings highlight LEOPARD's strong performance across various multimodal applications.

## 2 Related Work

**Text-rich MLLMs.** Text-rich images are traditionally processed in pipelines (Singh et al., 2019a; Hu et al., 2020), where an OCR module first recognized text from the image, followed by processing through a language

model. To improve efficiency and avoid error propagation, with the advent of MLLMs, end-to-end approaches become more popular recently. For instance, LLaVAR (Zhang et al., 2023) utilized a dataset of 400K instances with OCR-enhanced text to outperform LLaVA on various text-rich VQA tasks. Subsequent models such as UReader (Ye et al., 2023), TextMonkey (Liu et al., 2024c), and Mplug-DocOwl-1.5 (Hu et al., 2024a) recognized the importance of high-resolution encoding for accurate text comprehension, so they adopted strategies that cropped single images into multiple sub-images to preserve the original resolution during visual encoding. However, these approaches are primarily trained on single-image data, and struggle to generalize effectively to multi-image scenarios. Furthermore, the straightforward partitioning technique encounters challenges with multi-image inputs, as the sequence length rapidly increases with the number of images.

**Multi-image MLLMs.** Efforts have been made in training MLLMs with multi-image inputs due to the prevalence of multi-image scenarios in real-world applications. Mantis (Jiang et al., 2024) introduced a multi-image instruction tuning dataset on a variety of natural image scenarios. Besides, both VILA (Lin et al., 2023) and Idefics-2 (Laurençon et al., 2024b) incorporated image-text interleaved data during their pre-training. LLaVA-Next-Interleave (Li et al., 2024c) further extended this by incorporating videos and multi-view 3D data into the training pipeline. However, these works primarily target natural images and general visual understanding, leaving a gap in handling text-rich, multi-image scenarios. Natural images typically follow a different distribution from text-rich images and often do not demand high-resolution processing. As a result, many existing multi-image MLLMs struggle to generalize to text-rich scenarios. Our work aims to address this gap by specifically focusing on multi-image settings where text-rich images are the primary input. Very recently, multi-image training for MLLMs has attracted intense attention from researchers. Several concurrent efforts have included multi-image interleaved data to train their models, such as LLaVA-OneVision 08/2024 (Li et al., 2024b), Idefics3 (08/2024, (Laurençon et al., 2024a)), NVLM (09/2024, (Dai et al., 2024)), mPlug-DocOwl-2 (09/2024, (Hu et al., 2024b)), Molmo (09/2024, (Deitke et al., 2024)) and Qwen2-VL (09/2024, (Wang et al., 2024)). This trending paradigm highlights the significant practical value of multi-image MLLMs by enhancing their ability to tackle a wide range of real-world applications. The incorporation of multi-image instruction tuning data is therefore of paramount importance.

## 3 Method

LEOPARD follows the typical design of decoder-only vision language models (Liu et al., 2023b; 2024a; Li et al., 2024c), including a visual encoder, a vision language connector, and a language model (LM), as shown in Figure 2 (④⑤).

Specially, the input images are first passed through a visual encoder to extract high-level semantic features, which are then mapped into the language space via a vision-language connector. After this transformation, the visual tokens are interleaved with the textual tokens and fed into the language model, which processes them causally, leveraging text-visual context to generate coherent outputs that align with both modalities.

### 3.1 Multi-image Text-rich Instruction Dataset

To train LEOPARD, we create a large-scale instruction-tuning dataset named LEOPARD-INSTRUCT, comprising **925K** instances, with **739K** specifically tailored for text-rich, multi-image scenarios. Table 1 lists the composition of our data, with a detailed breakdown in the Appendix A.1.

In constructing LEOPARD-INSTRUCT, we surveyed existing open-source datasets but found only **154K** suitable instances containing text-rich, multi-image data. This volume was insufficient for effective instruction tuning, compared to prior MLLM studies (Jiang et al., 2024; Laurençon et al., 2024b; Li et al., 2024c). To address this data scarcity, we collect an additional **585K** high-quality instances of text-rich,

| Data Types | # Instances | Proportion |
|---|---|---|
| **Total Samples** | 925K | |
| Single-image | 186K | 20.10% |
| Multi-image | 739K | 79.89% |
| *Public | 154K | 16.65% |
| *New (Ours) | 585K | 63.24% |
| **Rationales** | | |
| *Existing | 214K | 23.14% |
| *New (Ours) | 250K | 27.02% |
| *None | 461K | 49.84% |
| **Domains** | | |
| Documents | 192K | 20.76% |
| Slide Decks | 16K | 1.73% |
| Tables | 48K | 5.19% |
| Charts | 353K | 38.16% |
| Webpages | 55K | 5.95% |
| Others | 261K | 28.22% |

Table 1: Data statistics of the LEOPARD-INSTRUCT dataset.

| Models | Visual Encoder | Resolution | Backbone LLM | Param. | PT. | IT. |
|---|---|---|---|---|---|---|
| Otter-9B (Li et al., 2023) | CLIP ViT-L | $224^2$ | LLaMA-7B | 9B | 30M | 5.1M |
| Emu2-Chat (Sun et al., 2023) | EVA-02-CLIP | $448^2$ | LLaMA-33B | 37B | - | 160M |
| MM1-7B-Chat (McKinzie et al., 2024) | CLIP ViT-H | $378^2$ | - | 7B | - | 1.5M |
| VILA1.5-8B (Lin et al., 2023) | SigLIP | $384^2$ | LLaMA3-8B | 8B | 50M | 1M |
| mPlug-DocOwl-1.5 (Hu et al., 2024a) | CLIP ViT-L | $448^2$ (x9 crops) | LLaMA-7B | 8B | 4M | 1M |
| Idefics2-8B (Laurençon et al., 2024b) | SigLIP | $980^2$ | Mistral-7B | 8B | 350M | 20M |
| LLaVA-NeXT-Inter (Li et al., 2024c) | SigLIP | AnyRes | Qwen1.5-7B | 7B | 1.3M | 1.2M |
| Llama-3.2-11B (Meta et al., 2024) | ViT-H/14 | $384^2$(x4 crops) | LLaMA3.1-8B | 11B | 6B | - |
| Qwen2-VL-7B (Wang et al., 2024) | CLIP ViT-bigG | Dynamic | Qwen2-7B | 7B | 800B Token | - |
| Mantis-LLaVA (Jiang et al., 2024) | SigLIP | $384^2$ | LLaMA3-8B | 8B | 0.5M | 1M |
| Mantis-Idefics2 (Jiang et al., 2024) | SigLIP | $980^2$ | Mistral-7B | 8B | 350M | 1M |
| LEOPARD-LLaVA (Ours) | SigLIP | Adapt HR. | LLaMA3.1-8B | 8B | 0.5M | 1.2M |
| LEOPARD-Idefics2 (Ours) | SigLIP | $980^2$ | Mistral-7B | 8B | 350M | 1.2M |
| LEOPARD-LLaVA-Pro (Ours) | SigLIP | Adapt HR. | LLaMA3.1-8B | 8B | 3.6M | 1.2M |

Table 2: A detailed comparison of the model training details between baseline models and LEOPARD, including image resolution, vision encoder, backbone LLM, number of parameters (Param.), pre-training (PT.) data size, and instruction tuning (IT.) data size of baselines. AnyRes denotes the resolution selecting method proposed by Liu et al. (2024a) and Adapt HR. represents the proposed adaptive high-resolution multi-image encoding strategy. We show the instance numbers (or token number if reported) for training of the models.

multi-image data. Each instance consists of a set of images paired with task instructions and responses. To collect these instances, we design multiple data pipelines, such as collecting multi-image data from the web, transforming textual data into multi-image data, and assembling multi-turn multi-image data. These methods are discussed in detail in the following paragraphs.

### 3.1.1 Newly Collected Multi-image Data (585K)

**Web Data Annotation**: We curate data from websites containing text-rich, multi-image content, such as articles featuring multiple charts or presentation slide decks. *Multiple charts* from the same article are often semantically interconnected, requiring sophisticated cross-image reasoning for comprehension. Currently, no datasets are specifically designed for multi-chart scenarios. To address this, we download charts from social reports of the Pew Research Center[1] which typically feature multiple, interrelated charts on the same topic. Following a common approach (Shi et al., 2024), we use GPT-4o to generate questions about these charts, as well as the corresponding answers. To further enhance LEOPARD's reasoning abilities, we prompt GPT-4o to create questions that necessitate cross-chart information synthesis and to provide step-by-step reasoning explanations before reaching the final answer. *Slide decks* offer another valuable source of text-rich, multi-image data, as combining information across multiple slides are often needed for complete understanding. We download slides from SlideShare[2], and again use GPT-4o to generate question-answer pairs, along with detailed reasoning steps. The GPT-4o annotation prompt is provided in the Appendix A.3. After manually reviewing a sample of 100 GPT-4o-annotated instances, we observed an accuracy rate exceeding 90%, indicating high-quality annotations.

**Transformed from Unimodal Textual Data**: Tables provide highly structured, organized quantitative information that requires both visual layout perception and textual understanding. However, existing vision-language tasks rarely address multi-table understanding. To fill this gap, we employ two strategies. First, we incorporate multi-table data from MultiHiertt (Zhao et al., 2022) and MultiTabQA (Pal et al., 2023) where tables are originally stored in JSON or DataFrame formats. We programmatically render tables as images to convert them into multimodal data. Details of the rendering process are in Appendix A.4. Second, we use single-table data from TableGPT (Li et al., 2024e) and split each table into sub-tables by randomly dividing rows and columns. These sub-tables are then rendered as individual images, creating multimodal, multi-table instruction data.

---

[1] https://www.pewresearch.org
[2] https://www.slideshare.net

| Dataset | Total | #Text | Domain | Anno. | Summary |
|---|---|---|---|---|---|
| Interleave (M4-Instruct) (Li et al., 2024c) | 500.8K | 21.3K | Natural images (real-world scenes, animals, landscapes, people, etc.) | N/A | Contains limited text-rich images; primarily focused on natural scenes. |
| Mantis-Instruct (Jiang et al., 2024) | 721K | 0 | Natural images (real-world scenes, animals, landscapes, people, etc.) | GPT-4V | No text-rich images included. |
| MP-DocStruct1M (Hu et al., 2024b) | 1.113M | 1.113M | Documents only | N/A | Contains only document images; single domain with low diversity. |
| MIMIC-IT (Li et al., 2023) | 2.8M | 0 | Natural images (real-world scenes, animals, landscapes, people, etc.) | N/A | No text-rich images included. |
| Leopard-Instruct | 739K | 739K | Text-rich images (documents, charts, tables, webpages, slide decks, etc.) | GPT-4o | Diverse text-rich images across various domains, with silver labels by GPT-4o. |

Table 3: Comparison of 5 multi-image instruction datasets. **Total** denotes the total number of multi-image samples and **# Text** is the text-rich samples number. **Anno.** denotes if the dataset has annotations.

**Assembled Multi-turn Multi-image Data**: Single-image datasets are significantly more abundant than multi-image resources. To leverage such data in training LEOPARD, inspired by Jiang et al. (2024), we randomly combine 2 to 4 single-image instances to create synthetic multi-image data. We first concatenate the selected images to form a multi-image input, then stack their corresponding question-answer pairs as multi-turn conversations. Prompts like "in the second image" and "from the image on the right-hand", are added to each turn to direct the model's focus to the appropriate image. These assembled samples enhance the model's ability to associate natural language references with corresponding visual features but also expand the model's exposure to diverse domains beyond the limited multi-image datasets. This assembly strategy is applied on single-image datasets including ArxivQA Li et al. (2024d), RICO Deka et al. (2017), FigureQA Kahou et al. (2018), and MapQA Chang et al. (2022).

**Augmenting with Rationales.** Unlike single-image tasks, multi-image scenarios often require MLLMs to aggregate information across multiple images. However, many existing datasets provide only final answers (Zheng et al., 2023; Hu et al., 2023), limiting the model's ability to learn cross-image reasoning skills. To address this limitation, we employ GPT-4o to generate chain-of-thought (CoT) rationales for multi-image datasets that lack such annotations. This results in 250K instances with GPT-annotated reasoning steps. (details in Appendix A.3)

### 3.1.2 Adopted Existing Data (340K)

We comprehensively investigate existing text-rich training resources and adopt the following data, including both multi-image and single-image instances from various domains. *Documents* are a common source of multi-image data, containing extensive text that requires cross-page context integration to fully capture the intended information. We include public multi-page document datasets (Tito et al., 2022; Landeghem et al., 2023; Zhu et al., 2022), covering a variety of document types such as scanned handwriting, printed documents, and digital PDFs. *Slide presentations*, similarly, include text-rich content spread across multiple pages. Besides downloading new data from the web, we use existing slides datasets Tanaka et al. (2023); Sefid et al. (2021). *Webpage snapshots* are composed of sequential images of webpages and provide visual context essential for interpreting user instructions on web-based tasks. We employ both web action prediction data (Mind2Web (Deng et al., 2023), OmniACT (Kapoor et al., 2024)) and web-based question answering data (WebScreenshots (Aydos, 2020), WebVision (Li et al., 2017), WebUI (Wu et al., 2023a)). *Data in other domains* are also included, such as infographics data (InfographicVQA Mathew et al. (2022)), mathematical diagrams (MathV360K Shi et al. (2024)), and abstractive diagrams (IconQA Lu et al. (2021)). We also incorporate some mixed-domain datasets that contain different types of text-rich images, including LLaVAR (Zhang et al., 2023),

Monkey Li et al. (2024g), and mPlugDocReason (Hu et al., 2024a). To preserve natural image understanding ability, we add samples from an instruction tuning dataset for natural images – ShareGPT4V (Chen et al., 2023). Finally, we remove duplicate instances from the whole LEOPARD-INSTRUCT dataset.

### 3.1.3 Comparison with Existing Datasets

We comprehensively compare several widely-used multi-image instruction-tuning datasets with LEOPARD-INSTRUCT in Table 3. Existing datasets such as Interleave (M4-Instruct), despite containing text-rich images, offer a very limited quantity (21.3K samples), insufficient for extensive training. While MP-DocStruct1M presents a substantial volume of document-focused data (1.113M samples), its scope remains narrow, limited exclusively to documents. Other datasets like Mantis-Instruct and MIMIC-IT primarily consist of natural images, limiting their applicability in text-rich scenarios. Addressing these limitations, our proposed LEOPARD-INSTRUCT dataset provides a robust collection of 739K text-rich images spanning diverse domains—including documents, charts, tables, webpages, and slide decks. This diversity and scale make LEOPARD-INSTRUCT particularly useful for training models to handle text-rich multi-image tasks.

## 3.2 Adaptive Resolution Multi-Image Encoding

Image resolution plays a critical role in MLLMs' visual comprehension, especially for text-rich images. Low resolutions can blur printed text, leading to perception errors and visual hallucinations. Existing MLLMs rely on pre-trained visual encoders typically capped at low resolutions like $224 \times 224$ or $336 \times 336$ pixels (Liu et al., 2023a; Lin et al., 2023; Jiang et al., 2024), limiting accurate textual understanding within images.

To address these limitations, a viable solution is to divide a high-resolution image into smaller sub-images, each processed independently by the model's visual encoder (Liu et al., 2024a; Dong et al., 2024). This partitioning captures finer visual details, enabling recognition of small or densely packed text. However, this approach greatly extends the visual feature sequence length, often surpassing the model's maximum sequence limit when handling multiple images. To mitigate this, we introduce an adaptive high-resolution multi-image encoding strategy.

**Image Allocation Computing**: To prevent the number of sub-image visual features from exceeding the LLM's maximum sequence length, we first set a budget $M$ for the total number of sub-images. We allocate this budget proportionally to each input image based on their original sizes. For each image **i** with dimensions $h_i \times w_i$, we calculate the initial number of sub-images $S_i$ as:

$$S_i = \left\lfloor \frac{h_i}{v} \right\rfloor \times \left\lfloor \frac{w_i}{v} \right\rfloor, \tag{1}$$

where $v$ is the resolution of visual encoder (e.g., $v = 364$ pixels). If the total number of patches satisfies $\sum_i S_i \leq M$, we proceed with these sub-image counts. Otherwise, we scale down these counts proportionally using a scaling factor $\alpha = \frac{M}{\sum_i S_i}$, resulting in adjusted sub-image counts:

$$S_i' = \lfloor \alpha S_i \rfloor . \tag{2}$$

**Image Partitioning**: For each image, we perform a grid search over possible number of rows $r$ and columns $c$ (where $1 \leq r, c \leq S_i'$ and $r \times c \leq S_i'$) to find the optimal cropping configuration that maximizes the effective resolution within the allocated sub-images (Li et al., 2024a). This configuration results in the original image being padded and resized to a target resolution of $(h_i' = r \times v, w_i' = c \times v)$. We then divide the image into $r \times c$ sub-images of size $(v \times v)$. Additionally, the original image is directly resized to $(v \times v)$, which provides a global view of the visual content.

**Image Encoding**: Most vision encoders transform an image into a sequence of visual features $\mathbf{v} \in \mathbb{R}^{L \times d}$, where $L$ denotes the sequence length and $d$ the feature dimension. Typically, $L$ is in the hundreds; for instance, the SigLIP encoder yields $L = 676$ and $d = 1152$ for an input image. Given that most LLMs have a sequence length limit of 8K tokens, this restricts the number of encoded images to at most 12 without

| Models | Text-Rich Multi-Image | | | | | |
|---|---|---|---|---|---|---|
| | MVQA$^D$ | DUDE | SlideVQA | MCQA | MH | Average |
| Otter-9B | 0.17 | 0.15 | 5.95 | 1.08 | 0.14 | 1.50 |
| Emu2-Chat | 17.58 | 13.79 | 0.60 | 2.40 | 0.72 | 7.02 |
| VILA-LLaMA3-8B | 30.75 | 19.75 | 24.72 | 1.87 | 3.66 | 16.15 |
| mPlug-DocOwl-1.5 | 35.85 | 16.94 | 4.54 | 0.26 | 0.86 | 11.69 |
| Idefics2-8B | 46.67 | 23.06 | 25.14 | 2.59 | 9.89 | 21.47 |
| LLaVA-NeXT-Inter | 39.92 | 24.04 | 23.46 | 14.34 | 3.55 | 21.06 |
| Llama-3.2-11B | 57.60 | 20.77 | 19.43 | 6.25 | 3.16 | 21.44 |
| Qwen2-VL-7B | **71.86** | 40.86 | 19.08 | 6.89 | 7.37 | 29.21 |
| Mantis-LLaVA | 31.89 | 17.73 | 16.81 | 9.72 | 3.46 | 15.92 |
| Mantis-Idefics2 | 51.61 | 27.74 | 24.02 | 12.97 | 5.48 | 24.36 |
| LEOPARD-LLaVA | 57.56 | 37.30 | 27.53 | 14.30 | 7.70 | 28.88 |
| LEOPARD-Idefics2 | 66.06 | 40.74 | 34.93 | 18.03 | 10.09 | 33.97 |
| LEOPARD-LLaVA-Pro | 70.60 | **43.82** | **37.51** | **24.30** | **12.12** | **37.67** |
| – compared to SoTA Qwen2-VL | (-1.3) | **(+3.0)** | **(+18.4)** | **(+17.4)** | **(+4.8)** | **(+8.5)** |

Table 4: Experiment results of baseline models and LEOPARD on 8 benchmarks of text-rich images. We use abbreviated benchmark names due to space limits. MVQA$^D$: Multi-page DocVQA, MCQA: MultiChartQA, MH: MultiHiertt. Following (Tito et al., 2022), for MVQA$^D$ and DUDE, we use average normalized Levenshtein similarity (ANLS) as the metric. For other benchmarks, accuracy is used as the metric, which measures if the predicted answer exactly matches any target answer. We highlight the best and the second best results with **Bold** and underline, respectively.

| Ablation Settings | Backbone | Adaptive | CoT | Text-Rich Multi-Image | | | | Text-Rich Single | | General | |
|---|---|---|---|---|---|---|---|---|---|---|---|
| | | | | MVQAv2 | DUDE | SlidesVQA | **Avg.** | TextVQA | DocVQA | MMMU | MathVista |
| LEOPARD-LLaVA | LLaMA-3.1 | ✓ | ✓ | 57.56 | 37.30 | 27.53 | **40.80** | 67.70 | 68.07 | 43.00 | 45.50 |
| – w/o Adaptive | LLaMA-3.1 | × | ✓ | 40.44 | 26.16 | 20.93 | 29.17(↓11.6) | 60.18 | 44.69 | 41.00 | 42.40 |
| – w/o CoT | LLaMA-3.1 | ✓ | × | 52.24 | 34.23 | 22.57 | 36.34(↓4.5) | 66.34 | 64.31 | 41.78 | 43.40 |
| – w/ LLaMA-3 | LLaMA-3 | ✓ | ✓ | 48.66 | 32.64 | 25.75 | 35.68(↓5.1) | 67.08 | 54.92 | 41.22 | 42.10 |

Table 5: Ablation study on LEOPARD-LLaVA: impact of removing or varying Adaptive and CoT components, and of using different backbones.

text input, severely limiting image capacity. To address this, inspired by pixel shuffling (Chen et al., 2024; Laurençon et al., 2024a), we concatenate $n$ adjacent visual features along the feature dimension, reducing $L$ by a factor of $n$. This results in a compressed visual feature sequence $\mathbf{v}' \in \mathbb{R}^{\frac{L}{n} \times nd}$, effectively allowing more images to be accommodated.

To incorporate these visual features into the LLM, we project the encoded visual feature sequences into the LLM's textual embedding space via a vision-language connector. To manage the variable lengths of feature sequences from partitioned images, we introduce special tokens to delineate image features, aiding in their distinction. Specifically, the sequence for the $i$-th image is formatted as: {Image $i$:  <Visual Feature Sequence> < /Img>}, where  and < /Img> are special tokens. An example of this formatting is shown in Figure 2.

## 4 Experiment

### 4.1 Implementation Details

**Model Architecture.** We train our models on two base architectures: LLaVA (Liu et al., 2023a) and Idefics2 (Laurençon et al., 2024b). For LEOPARD-LLaVA, we use SigLIP-SO-400M (Zhai et al., 2023) with $364 \times 364$ image resolutions as the visual encoder since it supports larger resolution than the commonly used $224 \times 224$ resolution CLIP visual encoder (Radford et al., 2021). Each image is encoded into a sequence of $26 \times 26 = 676$ visual features under a patch size of 14. With the visual feature pixel shuffling strategy, each

| Ablation Settings | Data size | Text-Rich Multi-Image | | | | Text-Rich Single | | General | |
|---|---|---|---|---|---|---|---|---|---|
| | | MVQA$^D$ | DUDE | SlidesVQA | **Multi Avg.** | TextVQA | DocVQA | MMMU | MathVista |
| Leopard-LLaVA | 925K | 57.56 | 37.30 | 27.53 | **40.80** | 67.70 | 68.07 | 43.00 | 45.50 |
| - w/o doc | 720K | 43.79 | 29.50 | 23.10 | 32.13(**8.7**↓) | 66.78 | 56.60 | 40.67 | 44.80 |
| - w/o chart | 524K | 54.33 | 35.65 | 18.73 | 36.23(**4.6**↓) | 66.86 | 50.78 | 41.89 | 39.60 |
| - w/o web | 870K | 54.62 | 35.70 | 20.79 | 37.02(**3.8**↓) | 67.40 | 67.82 | 41.78 | 44.00 |
| - only existing | 340K | 49.29 | 31.96 | 20.46 | 33.90(**6.9**↓) | 60.22 | 60.58 | 40.67 | 40.40 |

Table 6: Data ablation studies on Leopard-LLaVA: evaluating the impact of different data domains for instruction tuning, including doc , chart , and web , as well as using only existing data listed in Table 1.

| Models | General Vision-Language Tasks | | | | | Text-Rich Single | |
|---|---|---|---|---|---|---|---|
| | MIRB | MiBench | MMMU | MathVista | ScienceQA | TextVQA | DocVQA |
| Otter-9B | 20.74 | 43.72 | 30.89 | 22.00 | 60.44 | 23.18 | 3.53 |
| Emu2-Chat | 36.02 | 58.93 | 34.10 | 30.40 | 65.69 | 66.60 | 5.44 |
| VILA-LLaMA3-8B | 40.87 | 53.70 | 36.90 | 35.40 | 79.90 | 66.30 | 30.38 |
| mPlug-DocOwl-1.5 | 25.39 | 40.80 | 35.44 | 29.50 | 64.40 | 68.60 | 82.20 |
| Idefics2-8B | 33.02 | 46.39 | 42.90 | 45.00 | 89.04 | 70.40 | 67.30 |
| LLaVA-NeXT-Inter | **44.38** | **74.52** | 38.44 | 32.10 | 72.63 | 62.76 | 75.70 |
| Llama-3.2-11B | 20.96 | 34.33 | 50.70 | 51.50 | 71.99 | 73.60 | 88.40 |
| Qwen2-VL-7B | 59.96 | 52.91 | **54.10** | **58.20** | 78.38 | **84.30** | **94.50** |
| Mantis-LLaVA | 40.76 | 59.96 | 40.10 | 34.40 | 74.90 | 59.20 | 39.02 |
| Mantis-Idefics2 | 41.80 | 56.80 | 41.10 | 40.40 | 81.30 | 63.50 | 54.03 |
| Leopard-LLaVA | 42.00 | 60.80 | 43.00 | 45.50 | 85.57 | 67.70 | 68.07 |
| Leopard-Idefics2 | 41.38 | 61.74 | 40.11 | 44.80 | 90.38 | 80.40 | 74.79 |
| Leopard-LLaVA-Pro | 44.20 | 63.25 | 45.00 | 54.24 | **90.50** | 75.50 | 82.61 |

Table 7: Experimental results on out-of-domain evaluations, including general domain vision language tasks and text-rich single-image tasks.

image is further processed into a sequence of 169 visual features. We limit the maximum number of images ($M$) in each sample to 50, which produces up to $8,450$ visual features in total.

Following Liu et al. (2023a), we adopt a two-layer MLPs as the visual-language connector. We use LLaMA-3.1 (Meta et al., 2024) as the backbone language model.

For Leopard-Idefics2, we follow the architecture of Idefics2-8B which uses SigLIP-SO-400M as the visual encoder but increases its image resolution to $980 \times 980$ to make the text legible. The features outputted by the visual encoder are compressed with a feature resampler into 64 tokens per image. Idefics2-8B adopts the Mistral-7B (Jiang et al., 2023) as the LM.

**Training Details.** When training Leopard-LLaVA, we first train the visual-language connector using LLaVA's 558K multimodal pre-training dataset. Subsequently, we fine-tune the model (with both the connector and the LM unfrozen) using our Leopard-instruct data. As for Leopard-Idefics2, we directly finetune the pre-trained Idefics2 checkpoint on the Leopard-instruct dataset. To excel in text-rich scenarios, we additionally incorporate text-rich image data into the pre-training stage of Leopard-LLaVA, leading to the most functional model: Leopard-LLaVA-Pro. This includes Recap-COCO-30K (Li et al., 2024f), CC-3M (Changpinyo et al., 2021), Donut (Kim et al., 2022), the Cauldron (Laurençon et al., 2024b), and 4M Arxiv pages processed by an OCR toolkit[3]. We provide further training details in A.2.

---

[3] https://github.com/facebookresearch/nougat.

## 4.2   Baseline Models

We compare LEOPARD against a wide range of open-source MLLMs that support multi-image inputs. These baseline models include Otter-9B (Li et al., 2023), Emu2-Chat-34B (Sun et al., 2023), Mantis (Jiang et al., 2024), VILA (Lin et al., 2023), Idefics2-8B (Laurençon et al., 2024b), LLaVA-NeXT-Interleave (Li et al., 2024c), Qwen2-VL (Wang et al., 2024), and LLaMA3.2 (Meta et al., 2024). Models that only support a single image input are excluded from our comparisons, except for mPlug-DocOwl-1.5 (Hu et al., 2024a), as it is primarily trained on visual document data and demonstrates strong capabilities on text-rich image tasks. Table 2 demonstrates a detailed comparison of the model training details of between baseline models and our proposed LEOPARD, which highlights their architecture, image resolution and training data differences.

## 4.3   Evaluating Benchmarks

We evaluated LEOPARD and baseline methods across three categories of vision-language tasks on (1) single text-rich image evaluation, (2) multiple text-rich images evaluation, and (3) general reasoning evaluation. Benchmarks for (1) include TextVQA (Singh et al., 2019b) and DocVQA (Mathew et al., 2021). Benchmarks for (2) include Multi-page DocVQA (Tito et al., 2022), DUDE (Landeghem et al., 2023), SlideVQA (Tanaka et al., 2023), Multihiertt (Zhao et al., 2022), and MultiChartQA (Zhu et al., 2024), which cover a diverse range of typical multi-image tasks, such as document understanding and slide question answering. Benchmarks for (3) include MMMU (Yue et al., 2024), MathVista (Lu et al., 2024), ScienceQA (Saikh et al., 2022), MIRB (Zhao et al., 2024) and MiBench (Liu et al., 2024b), which evaluate MLLMs from different perspectives, including world knowledge, mathematics, *etc.*

## 4.4   Main Experimental Results

### *Question 1: How does* **Leopard** *compare to state-of-the-art MLLMs on vision-language tasks?*

LEOPARD achieves outstanding performance on **text-rich, multi-image** benchmarks, as shown in Table 4. Notably, both LEOPARD-LLaVA-Pro and LEOPARD-Idefics2 significantly outperform all baselines. LEOPARD-LLaVA-Pro becomes the strongest open-source MLLM in this area, achieving an average improvement of 8.46 points over the previous best.

In **single-image text-rich** scenarios shown in Table 7, LEOPARD outperforms several recent strong models, including VILA and LLaVA-NeXT. LEOPARD even achieves slightly higher average scores than the state-of-the-art mPlug model, despite mPlug being trained on 4M single-image data while LEOPARD is tuned on <200K. This demonstrates that training on multi-image data from LEOPARD-INSTRUCT also benefits model performance on single-image tasks.

In addition, we evaluate LEOPARD on **general-domain** benchmarks which contain both multi-image and single-image instances. As shown in Table 7, LEOPARD outperforms other open-source MLLMs on these benchmarks. Remarkably, LEOPARD surpasses Mantis, its counterpart multi-image model trained on the same foundational architecture and a comparable volume of data. This performance demonstrates the high quality and diversity of the LEOPARD-INSTRUCT dataset, which effectively preserves our model's general image understanding capabilities.

### *Question 2: Is the one-million text-rich multi-image dataset effective for instruction tuning?*

Mantis-Idefics2 is trained on a combination of natural *multi-image data* and *text-rich single-image* data. However, LEOPARD-Idefics2 outperforms Mantis-Idefics2 by 12.8 points on text-rich multi-image benchmarks. This disparity indicates that developing strong multi-image text-rich capabilities through cross-domain transfer, such as with Mantis data, presents significant challenges. This finding underscores the importance of optimizing LEOPARD using high-quality, diverse, and well-curated multi-image text-rich datasets that are specifically tailored for complex multi-image scenarios.

Furthermore, LEOPARD-Idefics2 surpasses its base model, Idefics2, by 6.4 points across three single-image text-rich benchmarks, though Idefics2 is trained on over 20M instruction data that includes text-rich tasks

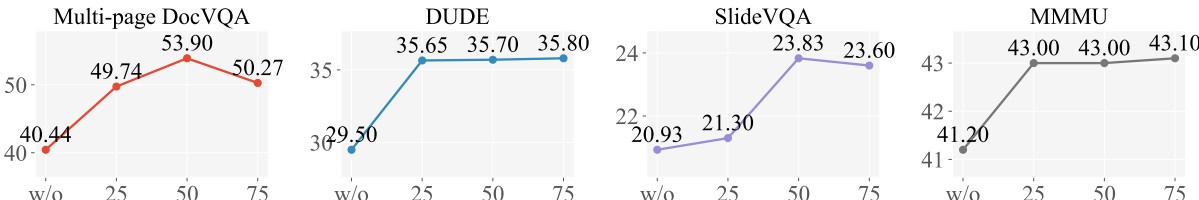

Figure 3: Impact of the sub-image budget $M$ on the resulting model across four benchmarks. *w/o* indicates no partitioning into sub-images.

like DocVQA and TextVQA. This highlights that the LEOPARD-INSTRUCT provides unique advantages to MLLMs that are not adequately addressed by existing datasets.

**Question 3: Does Adaptive high-resolution multi-image encoding improve MLLM performance?**

To assess the effectiveness of the proposed adaptive high-resolution multi-image encoding, we compared LEOPARD with a variant that excludes this feature (*i.e., w/o* Adaptive in Table 6). We notice a significant performance decline across all text-rich benchmarks, particularly on document-related benchmarks like DocVQA (-23.4), Multi-page DocVQA (-13.5), and DUDE (-9.8). This observation supports our hypothesis that high-resolution image encoding is especially beneficial for text-rich images, particularly with dense text content such as document pages.

### 4.5 More Analysis

**Question 4: How does data from different domains contribute to instruction tuning?**

LEOPARD-INSTRUCT mainly cover three main domains, *i.e.,* documents & slides ( doc ), tables & charts ( chart ), and websites ( web ). To assess the impact of data from different domains, we conduct ablation studies on three variants of LEOPARD, with the results presented in Table 6. Removing any part of the training data results in performance degradation. The most significant drop occurs when we exclude document data while removing web data leads to a slight decrease. Furthermore, eliminating all newly collected data (retaining only the existing 340K data) causes a substantial performance decline.

**Question 5: What is the influence of different image budgets in adaptive multi-image encoding?**

In our adaptive multi-image encoding module, we define a budget $M$ for the maximum number of sub-images that the model can process. To evaluate the impact of such image partitioning, we train LEOPARD using different values of $M$: 25, 50, 75, as well as a baseline setting where no image partitioning is applied and the number of sub-images equals the number of original images. According to the results plotted in Figure 3, model performance peaks or plateaus when $M$ is set around 50. Thus, we adopt 50 as the default value for training LEOPARD. These results show that increasing image numbers does not consistently improve performance, as input sequences can become excessively long and even exceed the model's sequence length limit.

**Question 6: How does the backbone language model affect the performance?**

To ensure a fair comparison with multi-image competitor models, Mantis-LLaVA and VILA1.5, we also evaluate a variant of LEOPARD using LLaMA-3 instead of LLaMA-3.1 , aligning its backbone language model architecture with these two baselines. According to Table 6, this substitution results in only a slight drop in average performance on text-rich multi-image tasks (2.2↓). Nevertheless, comparing with results in Table 4, LEOPARD-LLaMA-3 still substantially outperforms both baselines in all tasks, such as Multi-page DocVQA (+16.8 over Mantis and +17.9 over VILA) and DUDE (+14.9 over Mantis and +12.9 over VILA). These results indicate that LEOPARD's superior performance is not simply a result of the upgraded backbone large language models.

## 5 Conclusion

In this paper, we introduce LEOPARD, a novel MLLM specifically designed for text-rich, multi-image tasks. LEOPARD has two key innovations: (1) LEOPARD-INSTRUCT, a large-scale instruction-tuning dataset that encompasses a wide range of text-rich, multi-image instructions, and (2) an adaptive image encoding module capable of processing multiple high-resolution images efficiently. Our experimental results across diverse benchmarks highlight LEOPARD's superior performance compared to existing open-source MLLMs, particularly in text-rich multi-image scenarios. Further analysis and ablation studies underscore the effectiveness of both the collected dataset and adaptive encoding strategy, solidifying LEOPARD's contribution to multimodal research.

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

# A  Appendix

## A.1  Leopard-Instruct

To train LEOPARD, we created a large instruction-tuning dataset, LEOPARD-INSTRUCT, with 925K instances, including 739K designed for text-rich, multi-image scenarios. Despite surveying existing datasets, we found only 154K suitable text-rich, multi-image samples – insufficient for effective instruction tuning, which is far

from sufficient for effective instruction tuning, as shown in prior MLLM studies (Jiang et al., 2024; Laurençon et al., 2024b; Li et al., 2024c). To overcome this limitation, we developed several data collection pipelines to collect high-quality text-rich, multi-image data, resulting in additional 585K instances.

Table 8 provides a detailed breakdown of the composition of the LEOPARD-INSTRUCT dataset. This table includes the name, domain, and sample size of sub-datasets. Additionally, it specifies how we construct multi-image samples, the number of images per sample, and the presence of rationales.

| Dataset | Domain | Multi-image | Images | Rationales | #Samples (K) |
|---|---|---|---|---|---|
| ArxivQA (Li et al., 2024d) | Doc | Reformed | 1-3 | Existing | 81 |
| DUDE (Landeghem et al., 2023) | Doc | Public | 1-50 | Augmented | 23 |
| MP-DocVQA (Tito et al., 2022) | Doc | Public | 1-20 | Augmented | 36 |
| DocVQA (Mathew et al., 2021) | Doc | No | 1 | None | 39 |
| TAT-DQA (Zhu et al., 2022) | Doc | Reformed | 2-5 | Augmented | 13 |
| SlidesGeneration (Sefid et al., 2021) | Slides | Repurposed | 1-20 | Augmented | 3 |
| SlidesVQA (Tanaka et al., 2023) | Slides | Public | 20 | Augmented | 10 |
| Slideshare | Slides | Collected | 2-8 | Augmented | 3 |
| Multihiertt (Zhao et al., 2022) | Table | Public | 3-7 | Existing/Augmented | 15 |
| MultiTabQA (Pal et al., 2023) | Table | Public | 1-2 | Augmented | 6 |
| TableGPT (Li et al., 2024e) | Table | Split | 2 | Existing | 4 |
| TabMWP (Lu et al., 2023) | Table | No | 1 | Existing | 23 |
| ChartGemma (Masry et al., 2024) | Chart | Reformed | 1-4 | Existing | 65 |
| DVQA (Kafle et al., 2018) | Chart | Reformed | 1-3 | None | 200 |
| FigureQA (Kahou et al., 2018) | Chart | Reformed | 1-2 | None | 36 |
| ChartQA (Masry et al., 2022) | Chart | Reformed | 2 | Augmented | 32 |
| Pew_MultiChart | Chart | Collected | 2 | Augmented | 20 |
| Mind2Web (Deng et al., 2023) | Web | Split | 1-5 | None | 7 |
| WebsiteScreenshots (Aydos, 2020) | Web | No | 1 | Augmented | 2 |
| Omniact (Kapoor et al., 2024) | Web | No | 1 | None | 1 |
| RICO (Hsiao et al., 2024) | Web | Reformed | 1-4 | None | 25 |
| WebVision (Li et al., 2017) | Web | No | 1 | Existing | 1 |
| WebUI (Wu et al., 2023a) | Web | No | 1 | None | 19 |
| LLaVAR (Zhang et al., 2023) | Mix | No | 1 | Existing | 15 |
| MathV360k (Shi et al., 2024) | Mix | No | 1 | None | 38 |
| Monkey (Li et al., 2024g) | Mix | Reformed | 1-3 | None | 92 |
| MPlugDocReason (Hu et al., 2024a) | Mix | No | 1 | Existing | 25 |
| IconQA (Lu et al., 2021) | Other | Public | 1-6 | Augmented | 64 |
| InfographicVQA (Mathew et al., 2022) | Other | No | 1 | Augmented | 23 |
| MapQA (Chang et al., 2022) | Other | Reformed | 1-2 | None | 4 |
| Total | - | - | - | - | 925 |

Table 8: Details of the constructed LEOPARD-INSTRUCT dataset. Images denotes the image number of one sample in each dataset.

We draw a chart to illustrate the data composition of LEOPARD-INSTRUCT dataset 4.

## A.2 Training Details

For LEOPARD-Idefics2, we note that the Idefics2 model is pre-trained on a dataset comprised of over 350M multimodal samples. Given the computational challenges of reproducing such extensive pre-training, and to ensure a fair comparison with baselines that utilize the pre-trained Idefics2 checkpoint, we directly adopt Idefics2' visual feature resampler and fine-tune the model on the LEOPARD-INSTRUCT dataset.

We train both LEOPARD-LLaVA and LEOPARD-Idefics2 on 64 A100-40G GPUs with a global batch size of 128. We use the AdamW optimizer with $\beta_1 = 0.9$, $\beta_2 = 0.999$. Following (Jiang et al., 2024), we use a learning rate of $1 \times 10^{-5}$ for LEOPARD-LLaVA and $5 \times 10^{-6}$ for LEOPARD-Idefics2 to protect its pretrian knowledge. We use a cosine learning rate scheduler with a linear learning rate warm-up for the first 3% steps.

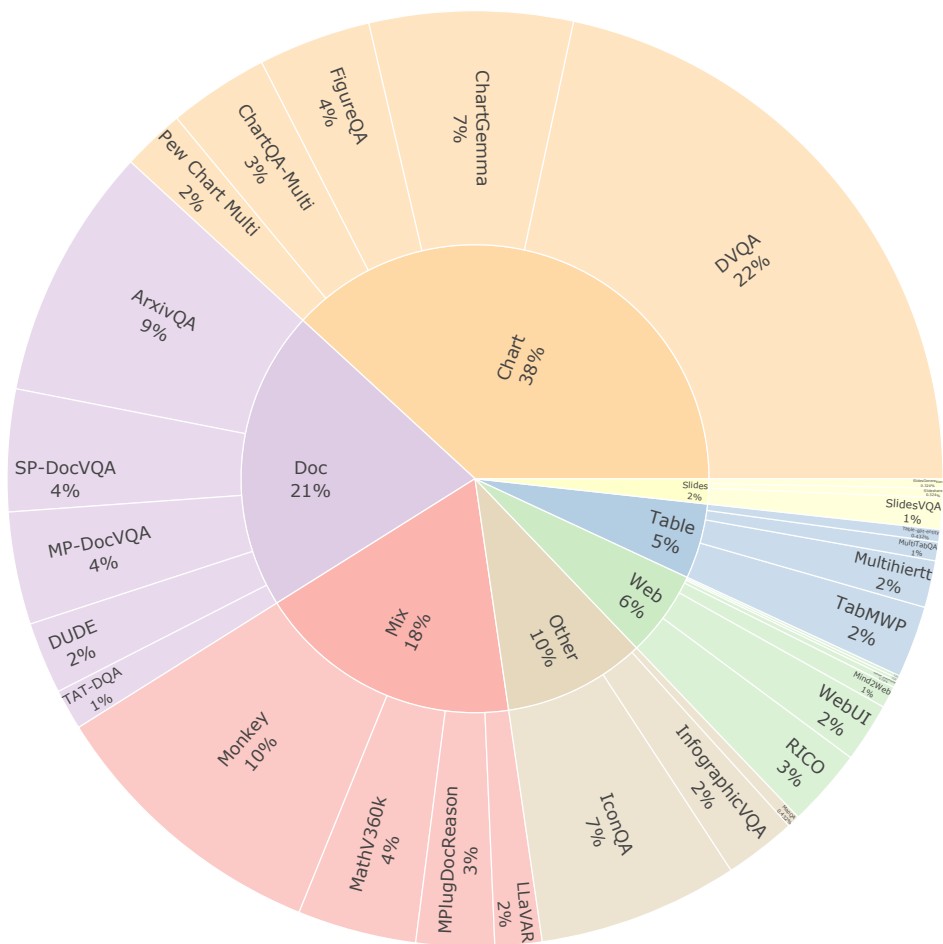

Figure 4: An illustration of the proportion of sub-datasets and domains in the proposed dataset.

All model variants are trained 1 epoch under the same hyperparameters. It takes around 120 GPU days to train LEOPARD under both settings.

### A.3 Prompts

We specify the prompt used during the data construction process as follows:

---

**Slides Q-A Generation Prompt**

You are given a set of images from a slides. Please generate 10 meaningful and distinct questions about the content of the slides.

You are supposed to generate the questions, the answers, and detailed explanations for the answers.
The questions should be clear, concise, and straightforward. The answers should be a few words or phrases.

You should ask questions about the details of the slides, including the tilte, the authors, and the figures and tables on the slides.

The output format should be in JSON format, with the following structure:
 [{"Question_0":"...","Answer_0":"...","Rationale_0":"..."},
{"Question_1":"...","Answer_1":"...","Rationale_1":"..."}, ...]

---

Figure 5: The prompt used for generating Q-A pairs with rationales for slide decks data.

---

**Webpage Q-A Generation Prompt**

You are given a screenshot of a website. Please generate 10 meaningful and distinct questions about the screenshot. You should pay attention to the textual content, the layout, and the elements on the web screenshot.

You are supposed to generate the questions, the answers, and detailed explanations for the answers. The questions should be clear, concise, and straightforward. The answers should be a few words or phrases.

You should ask questions about the webpage description, the elements on the webpage, and the uses of buttons on the webpage.

The output format should be in JSON format, with the following structure:
[{"Question_0":"...","Answer_0":"...","Rationale_0":"..."},
{"Question_1":"...","Answer_1":"...","Rationale_1":"..."}, ...]

---

Figure 6: The prompt used for generating Q-A pairs with rationales for webpage data.

---

**Rationale Augmentation Prompt**

You are an expert in multi-page visual questions.
Based on the following question and answer, please generate a rationale that derives the answer.
### Question: {question}
### Answer: {answer}
### Rationale:

---

Figure 7: We use this prompt for the generation of chain-of-thought rationales given original question, answer, and images.

### A.4 Details of Table Rendering

To convert the textual table dataset into a multimodal dataset, the JSON or DataFrame format data is transformed into tabular images using Python. We utilize three Python packages, *i.e.,* dataframe_image[4],

---

[4] https://github.com/dexplo/dataframe_image.

pandas[5], and matplotlib[6] with various styling to enhance the diversity of the rendered images. To ensure the clarity and legibility of the plotted images, the original data is filtered by excluding any tables that contain more than 20 rows. This threshold was set to maintain the recognizability of the resulting images.

## A.5 Qualitative Results

We show two examples to give an illustrative demonstration of the model's performance. As can be seen from Figure 8, LEOPARD can not only capture detailed data in multiple tables precisely but also perform cross-table calculations, therefore it can answer the complex question correctly. Another example is demonstrated in Figure 9. LEOPARD can accurately perceive the prominent information under a high-resolution four-page document, demonstration effective text-rich abilities under multi-image scenarios.

---

[5]https://pandas.pydata.org/.
[6]https://matplotlib.org/.

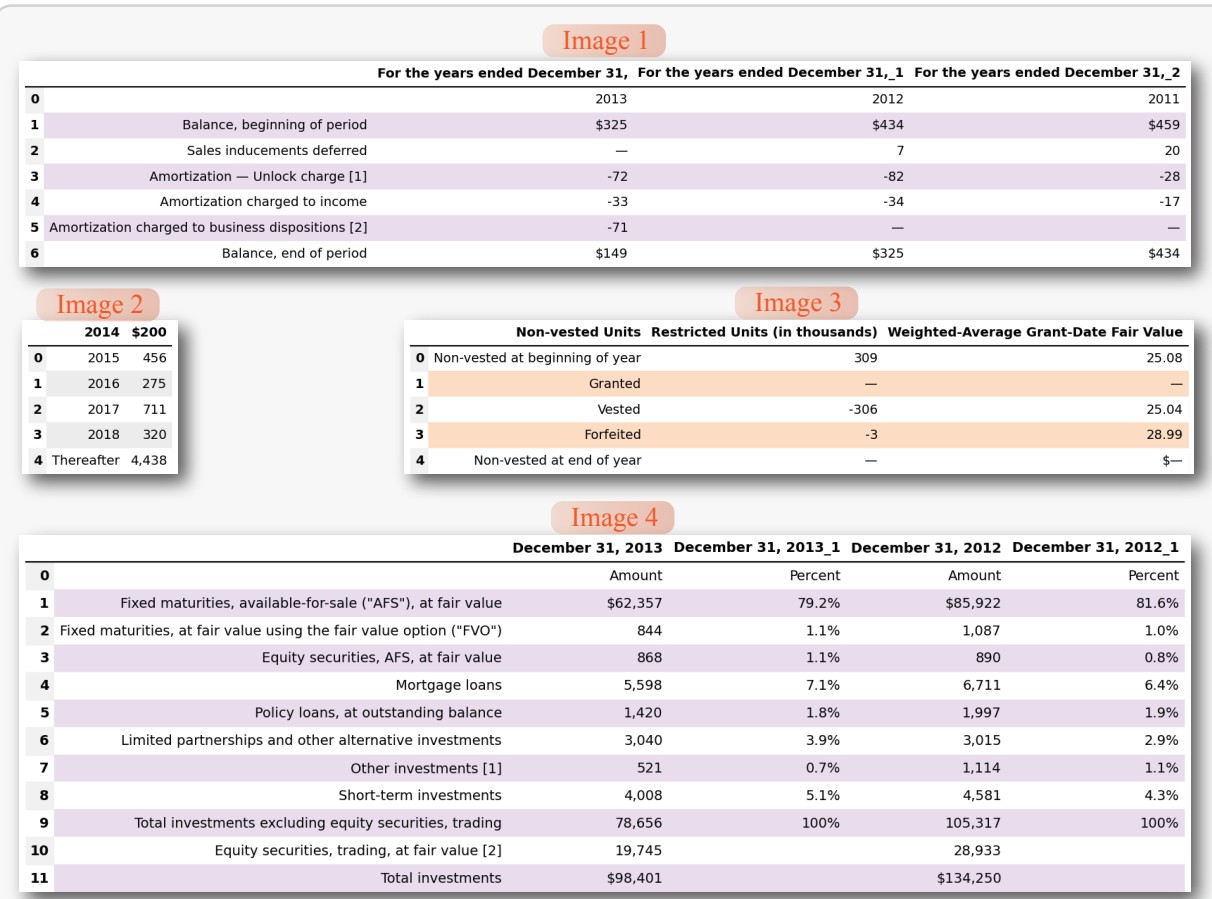

**Question**: What is the average value of Fixed maturities, available-for-sale ("AFS"), at fair value for Amount and Balance, beginning of period in 2013 ?

**LEOPARD**: To find the average value of Fixed maturities, available-for-sale ("AFS"), at fair value and Balance, beginning of period in 2013, we need to follow these steps:

1. Identify the values from the images:
   - Fixed maturities, available-for-sale ("AFS"), at fair value for Amount in 2013: $62,357 (from the fourth image).
   - Balance, beginning of period in 2013: $325 (from the first image).

2. Add these two values together:
   $[62{,}357 + 325 = 62{,}682]$

3. Divide the sum by 2 to find the average:
   $[\frac{62{,}682}{2} = 31{,}341]$

Thus, the average value of Fixed maturities, available-for-sale ("AFS"), at fair value and Balance, beginning of period in 2013 is $31,341.

Answer: 31341.0.

Figure 8: An example of multi-table reasoning of LEOPARD.

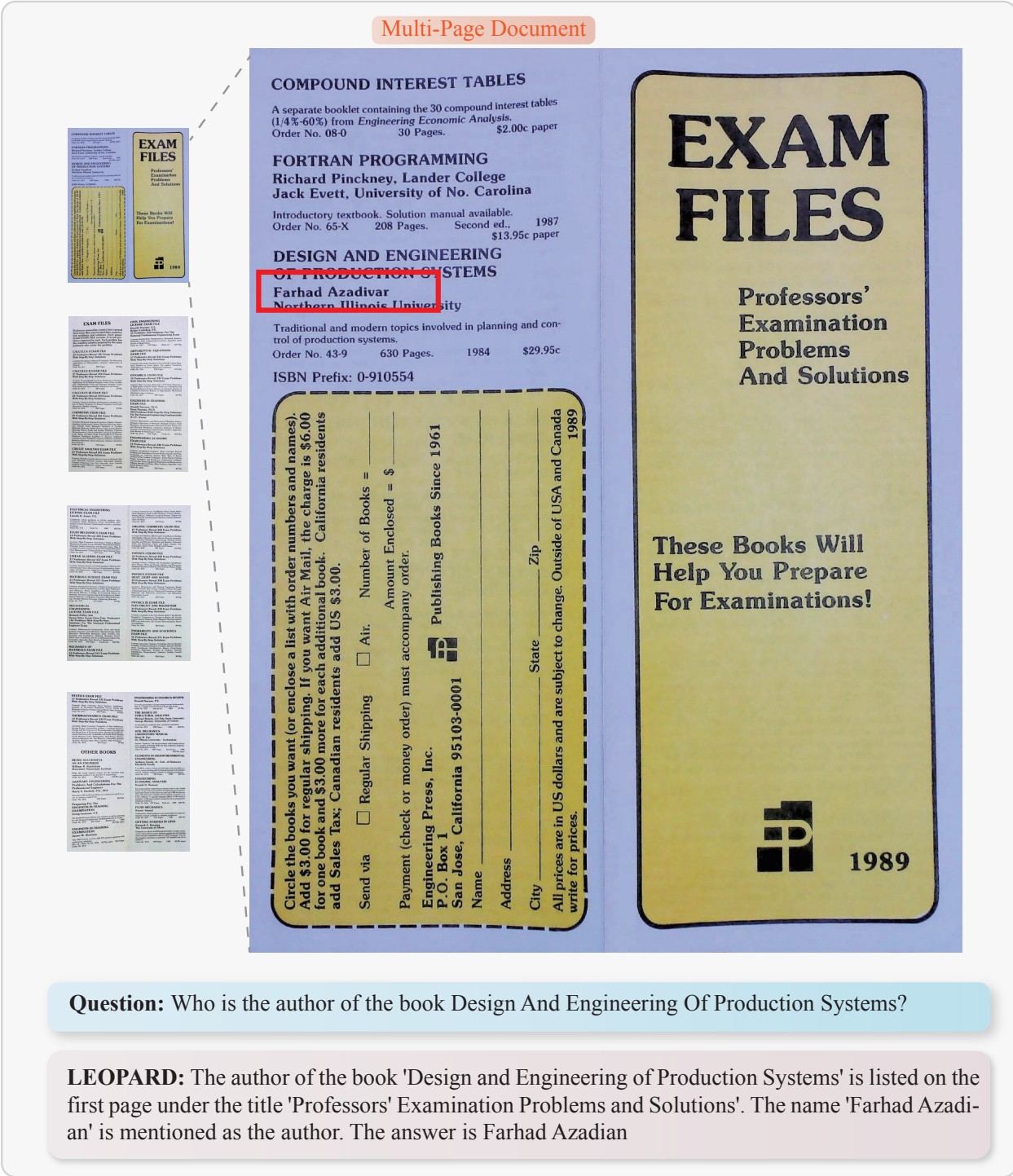

Figure 9: An example of multi-page document question answering of LEOPARD.

