# OpenReview forum: "Leopard: A Vision Language Model for Text-Rich Multi- Image Tasks"
_TMLR — Accepted by TMLR_

### Review · Reviewer_eP3s · 2025-04-01

**Summary Of Contributions:**

This paper introduces Leopard, a vision language model specifically designed for text-rich multi-image tasks. The key contributions include:

- Creation of a large-scale instruction-tuning dataset (approximately 1 million instances) specifically tailored for text-rich, multi-image scenarios. This dataset was developed through multiple approaches including web data collection, transformation of unimodal text data, and reformatting single-image resources.
- Development of an adaptive high-resolution multi-image encoding module that dynamically allocates visual sequence length based on image aspect ratios and resolutions, addressing the challenge of balancing image resolution with sequence length limitations.
- Comprehensive evaluation across 12 vision-language benchmarks showing that Leopard consistently outperforms state-of-the-art models (like Llama-3.2 and Qwen2-VL) on text-rich multi-image tasks while maintaining competitive performance on single-image and general tasks.

**Audience:**

Yes

**Broader Impact Concerns:**

This paper does not raise significant ethical concerns. The authors do not discuss potential misuse scenarios or data biases, which could be worth addressing, but these concerns are not critical given the application domain.

**Claims And Evidence:**

Yes

**Requested Changes:**

Clarify the distinction between "fully open-sourced training instances" and the GPT-4o annotation process used for data augmentation

**Strengths And Weaknesses:**

Strengths

- Addresses a significant gap in current MLLMs by focusing on text-rich multi-image scenarios (documents, presentations, webpage sequences)
- The dataset creation process is comprehensive and well-documented
- The adaptive high-resolution encoding approach effectively handles varying image resolutions while managing sequence length constraints
Extensive evaluation with detailed ablation studies supports the importance of both the dataset and encoding method

Weaknesses

Some techniques, like pixel shuffling for compressing visual feature sequences, build on existing approaches rather than introducing entirely novel methods
Several concurrent efforts in multi-image training for MLLMs were released around the same time (August-September 2024), somewhat reducing the novelty

---

> ### Author Response · Authors · 2025-04-23
>
> ## Techniques build on existing approaches rather than introducing entirely novel methods
>
> Pixel shuffling for compressing visual feature sequences has been widely adopted in prior work [1][2] due to its simplicity and effectiveness. We selected these components based on their strong empirical performance and practical utility. However, without our dataset and Adaptive Resolution Encoding strategy, models such as LLaVA, Idefics, and other approaches still struggle in challenging, text-rich scenarios.
>
> Our contributions focus on enhancing vision-language models’ ability to understand text-rich images, including documents, slide decks, charts, and webpages. In addition to the dataset construction, our Adaptive Resolution Encoding strategy is specifically designed to handle multi-image inputs with varying resolutions under a unified sequence length constraint. Unlike Qwen2-VL’s naïve dynamic resolution approach—which resizes each image independently without considering the global token budget—our method adaptively allocates visual tokens across all images, preserving more fine-grained details in text-dense inputs.
>
> [1] Expanding Performance Boundaries of Open-Source Multimodal Models with Model, Data, and Test-Time Scaling
>
> [2] Eagle: Exploring The Design Space for Multi- modal LLMs with Mixture of Encoders
>
>
> ## Clarify the distinction between "fully open-sourced training instances" and the GPT-4o annotation process used for data augmentation
>
> We appreciate the reviewer’s comment and apologize for the potential misunderstanding. We use the term “fully open-sourced training instances” to emphasize that the entire dataset (including images, instructions, rationales, and answers) is publicly released. While we use GPT-4o for data augmentation, this is consistent with common practice in many open-source datasets (e.g., MathV360K [1], ShareGPT4V [2], Mulberry [3]), where closed-source models are used for annotation but the resulting data is open-sourced. We have also released the exact prompts used in data generation (Appendix A.3) to ensure transparency. To avoid confusion, we will revise the term in the manuscript for clarity. Specifically, we changed the sentence:
>
> “Remarkably, our approach achieves outstanding performance using only 1.2M fully open-sourced training instances, outperforming models that rely on large-scale in-house data, highlighting its efficiency and effectiveness.”
>
> to:
>
> “Remarkably, our approach achieves outstanding performance using only 1.2M training instances, all of which are fully open-sourced, demonstrating both high efficiency and effectiveness compared to models trained on large-scale in-house data.”
>
>
> [1] Math-llava: Bootstrapping mathematical reasoning for multimodal large language models
>
> [2] Sharegpt4v: Improving large multi-modal models with better captions
>
> [3] Mulberry: Empowering MLLM with o1-like Reasoning and Reflection via Collective Monte Carlo Tree Search

---

### Review · Reviewer_k7Dh · 2025-04-13

**Summary Of Contributions:**

This paper presents *Leopard*, a multimodal large language model tailored for tasks involving multiple text-rich images. The authors contribute a high-quality, 1M-scale instruction-tuning dataset and propose an adaptive encoding module that optimizes visual token allocation based on image properties. The model achieves strong performance on challenging benchmarks, surpassing state-of-the-art baselines with only open-sourced data.

**Audience:**

Yes

**Broader Impact Concerns:**

If the collected data is compliant and can be open-sourced, it would make a significant contribution to the field.

**Claims And Evidence:**

Yes

**Requested Changes:**

See the weakness

**Strengths And Weaknesses:**

## Strengths

1. The dataset is large and, if open-sourced, would be a valuable contribution to the community.

2. The model demonstrates strong performance, surpassing state-of-the-art models.

## Weaknesses

1. I recommend a clearer comparison with existing datasets such as Interleave to better highlight the value of the newly collected dataset.

2. The rationale behind the Adaptive Resolution Multi-Image Encoding module is unclear—why not adopt the naïve resolution approach used in Qwen2-VL?

3. Table 4 needs further refinement, as the current version is difficult to understand.

---

> ### Author Response · Authors · 2025-04-23
>
> ## Clearer comparison with existing datasets
>
> Thanks for the helpful suggestion. In the below table and the revised version (see Table 3 and Section 3.1.3), we added a comparison with existing datasets. From the table we can see, Interleave (M4-Instruct) has text-rich images but only 21.3K samples, which isn’t enough for large-scale training. MP-DocStruct1M has more data (1.1M samples) but is limited to documents. Others like Mantis-Instruct and MIMIC-IT focus on natural images. To address this, our Leopard-instruct dataset includes 739K text-rich images across diverse formats—documents, charts, tables, webpages, slides—making it more suitable for training on complex, text-rich multi-image tasks.
>
> | Dataset                | Total Multi-image Samples | # Text-rich Images | Domain                                                                 | Annotation | Summary                                                                                          |
> |------------------------|---------------------------|--------------------|------------------------------------------------------------------------|------------|--------------------------------------------------------------------------------------------------|
> | Interleave (M4-Instruct) [1] | 500.8K                   | 21.3K              | Natural images (real-world scenes, animals, landscapes, people, etc.)  | N/A        | Contains limited text-rich images; primarily focused on natural scenes.                          |
> | Mantis‑Instruct [2]         | 721K                     | 0                  | Natural images (real-world scenes, animals, landscapes, people, etc.)  | GPT‑4V     | No text-rich images included.                                                                    |
> | MP‑DocStruct1M [3]          | 1.113M                   | 1.113M             | Documents only                                                         | N/A        | Contains only document images; single domain with low diversity.                                 |
> | MIMIC‑IT [4]                | 2.8M                     | 0                  | Natural images (real-world scenes, animals, landscapes, people, etc.)  | N/A        | No text-rich images included.                                                                    |
> | Leopard‑Instruct            | 739K                     | 739K               | Text-rich Images (document, charts, tables, webpages, slide decks etc.)| GPT-4o     | Diverse text-rich images across various domains, with silver labels by GPT-4o.                   |
>
> [1] LLaVA-NeXT-Interleave: Tackling Multi-image, Video, and 3D in Large Multimodal Models
>
> [2] MANTIS: Interleaved Multi-Image Instruction Tuning
>
> [3] mPLUG-DocOwl2: High-Resolution Compressing for OCR-Free Multi-Page Document Understanding
>
> [4] Otter: A Multi-Modal Model with In-Context Instruction Tuning
>
>
> ## Rationale behind the Adaptive Resolution Multi-Image Encoding module
>
> We thank the reviewer for raising this important question. Qwen2-VL adopts a naïve dynamic resolution strategy by resizing each image to fit within a predefined fixed maximum pixel limit. While this helps control the token length per image, it does not take into account the total number of images or resolution differences among them.
> As a result, two issues arise: (1) the total token usage may exceed the global budget when multiple images are present, leading to potential truncation or failure;; (2) images with high native resolution may be overly downsampled even when the budget allows for higher fidelity.
> In contrast, our Adaptive Resolution Multi-Image Encoding method allocates resolution across images based on a global token budget. It takes into account the varying resolutions of images and adjusts the resizing plan accordingly. This enables more efficient utilization of the token budget and improves performance on text-rich, multi-image tasks.
>
>
> ## Table 4 needs further refinement
>
> Thank you for the suggestion. We have carefully revised Table 4 to enhance its clarity. Kindly refer to the updated version in the revised PDF.

---

> > ### Author Response · Authors · 2025-04-23
> > **Image encoding comparison**
> >
> > We conduct an image encoding experiment to compare Qwen2-VL's naive dynamic resolution with the proposed adaptive resolution multi-image encoding.
> >
> > ### Experiment Setup
> > We compute visual tokens by first splitting each resized image into 14×14 patches and then compressing every 2×2 group of patches into a single feature, with two additional tokens for <|vision_start|> and <|vision_end|>, yielding
> > visual feature number = $\\frac{h'}{28}\times\frac{w'}{28} + 2$
> >
> > Under Naïve Dynamic Resolution (from Qwen2-VL), each image is independently resized to satisfy fixed pixel thresholds (min⁡_pixels=4×28^2=3,136, max⁡_pixels=16,384×28^2=12,845,056).
> >
> > +Equal-Split is obtained by taking Qwen2-VL’s total maximum pixel budget (12,845,056), dividing it equally among all input images, and then applying the per-image resizing rules to compute tokens—thereby avoiding global truncation.
> >
> > Under Adaptive Resolution Encoding, we enforce a global token budget of 50×169=8450 across all images: each image’s raw resolution is used to calculate its proportional share of the 8450 tokens, and then each image is resized to its allocated budget.
> > Assume the maximum model sequence length is 16000.
> >
> > ### Representative Scenarios & Token Allocation
> >
> >
> > | Scenario | Image # | Original Resolution | Naïve Dynamic Resolution Tokens | +Equal-Split | Adaptive Resolution Tokens |
> > |----------|---------|---------------------|----------------------------------|--------------|-----------------------------|
> > | A        | 1       | 8000×6000           | 16 000 (truncated)              | 8114         | 7754                        |
> > |         | 2       | 1000×1000           | 0 (truncated)                   | 1298         | 627                         |
> > |         | Total   |                     | **16000**                      | **9412**     | **8381**                    |
> > | B        | 1       | 5000×4000           | 16 000 (truncated)              | 2670         | 3382                        |
> > |         | 2       | 4000×3000           | 0 (truncated)                   | 2702         | 3187                        |
> > |         | 3       | 3000×2000           | 0 (truncated)                   | 2648         | 1538                        |
> > |         | 4       | 512×512             | 0 (truncated)                   | 326          | 66                          |
> > |         | 5       | 512×512             | 0 (truncated)                   | 326          | 66                          |
> > |         | 6       | 512×512             | 0 (truncated)                   | 326          | 66                          |
> > |        | Total   |                     | **16000**                      | **8 998**    | **8 305**                   |
> >
> > ### Observision
> >
> > From the table, it is clear that Naïve Dynamic Resolution immediately exhausts the token cap (16000) on the first large image in both scenarios, leaving all subsequent images completely unencoded. Equal-Split Resolution bounds the total token count by equally dividing pixel capacity, but this treats every image identically—small images receive the same budget as massive ones (a 5000×4000 image and a 3000×2000 image yield similar tokens in scenario B). In contrast, Adaptive Resolution Encoding always stays within the strict token budget and distributes tokens proportionally to each image’s raw demand, ensuring that both large and smaller images retain appropriate resolution without truncation.

---

### Review · Reviewer_W7yP · 2025-04-15

**Summary Of Contributions:**

This paper introduces Leopard, a multimodal large language model (MLLM) designed to handle multiple text-rich images, addressing two key challenges: insufficient training data and balancing resolution with sequence length. The authors develop a large-scale instruction dataset (925K samples) and an adaptive high-resolution encoding mechanism that dynamically allocates visual tokens based on image properties. Experiments show Leopard significantly outperforms existing models on text-rich multi-image tasks while maintaining competitive performance on single-image benchmarks.

**Audience:**

Yes

**Claims And Evidence:**

Yes

**Requested Changes:**

See weaknesses

## Additional Questions
1. Have you tried integrating large-scale OCR data in the pre-training process? I am wondering whether that will help the MLLM do better.

**Strengths And Weaknesses:**

## Strengths
1. The authors curated a comprehensive dataset (Leopard-instruct) comprising 739K multi-image and 186K single-image samples across diverse domains. They curate data from web, and transformed single-image data into multi-image data. They also synthesized inst data using powerful LLMs like GPT-4o
2. The adaptive high-resolution encoding module addresses sequence length limitations by allocating visual tokens based on image properties. As shown in Figure 3, increasing the sub-image budget to around 50 optimally balances resolution and sequence length, improving performance on Multi-page DocVQA from 40.44 to 53.90.
3. Leopard-LLaVA-Pro outperforms Qwen2-VL by an average of 8.5 points across text-rich multi-image benchmarks (Table 3), with particularly significant gains on SlideVQA (+18.4) and MultiChartQA (+17.4).
4. The authors thoroughly analyze component contributions, showing that removing adaptive high-resolution encoding causes performance to drop by 11.6 points on text-rich multi-image tasks (Table 4, p.8), confirming its critical role.

## Weaknesses
1. One possible missing comparison may be the OCR+ LLM (LLaMA-3.1-8B) to how much better Leopard is compared to this simple baseline. This helps people better position the test-rich MLLM.
2. One minor concern is that the main novelty of this paper is the data construction, lacking new designs on model arch. I do think this is a minor issue that does not affects the paper's acceptance. But it worth for the authors to consider the potential better MLLM arch design for text-rich domain.

---

> ### Author Response · Authors · 2025-04-23
>
> ## Missing comparison of OCR+ LLM (LLaMA-3.1-8B)
>
> We thank the reviewer for the helpful suggestion. Following it, we added an LLM+OCR baseline using the Tesseract OCR engine (v5.5.0, released on November 10, 2024) and LLaMA 3.1. The results are included in the updated table above.
>
> Our model (Leopard-LLaVA-8B) is also based on LLaMA 3.1, making the comparison fair. We observe that the performance of LLaMA-3.1-8B + OCR is significantly worse than our Leopard. There are two main reasons: (1) When the layout of some pages that includes charts and slide decks is complex, OCR often fails to extract the structural information embedded in the images. (2) In some scanned documents, handwritten text negatively impacts OCR accuracy.
>
>
> | Models                  | MVQA^D | DUDE  | SlideVQA | MCQA  | MH    | Average |
> |-------------------------|--------|-------|----------|-------|-------|---------|
> | LLaMA-3.1-8B + OCR       | 28.36  | 15.49 | 19.92    | 0.52  | 0.72  | 13.00   |
> | Leopard-LLaVA-8B        | 57.56  | 37.30 | 27.53    | 14.30 | 7.70  | 28.88   |
> | Leopard-LLaVA-8B-Pro    | 70.60  | 43.82 | 37.51    | 24.30 | 12.12 | 37.67   |
>
>
>
> ## Lacking new designs on model architecture
>
> We thank the reviewer for recognizing the novelty of our data construction. The full dataset and code have been open-sourced to support future research.
>
> While our model follows the standard LLaVA-style architecture, it has target adaptations to handle text-rich, multi-image scenarios. In particular, we propose an Adaptive Resolution Multi-Image Encoding strategy, which improves over both fixed-resolution approaches (e.g., LLaVA) and the naïve dynamic resolution used in Qwen2-VL.
>
> Specifically, Qwen2-VL resizes images based on predefined **per-image minimum and maximum pixel limits**. If the limits are set too low, the resulting image resolution is poor; if set too high, the token length becomes excessively long.
>
> In contrast, our method **globally allocates the visual token budget across all input images**, taking into account each image’s resolution and the overall sequence length constraint. This approach enables a more balanced and effective encoding of images with varying sizes. Our experiments show that this adaptation is crucial for improving model performance in the target domain, with an average performance drop of 11.6 points observed when adaptive resolution encoding is removed in text-rich multi-image evaluations.
>
>
>
>
> ## Integrating large-scale OCR data in the pre-training process
>
> We thank the reviewer for their attention to technical details and for raising this insightful question. Yes, we have incorporated OCR-related data into the pre-training phase to enhance the model’s performance in text-rich scenarios. As mentioned in the Training Details section of the experiments, we introduce additional text-rich image datasets during the pre-training of Leopard-LLaVA, resulting in our strongest model, Leopard-LLaVA-Pro. On top of the original LLaVA pre-training, we further utilize Recap-COCO-30K [1], CC-3M [2], and Donut [3]. Among them, the Donut dataset contains a large amount of OCR-style training data. After incorporating this additional OCR-related data, Leopard-LLaVA-Pro shows significant performance gains compared to Leopard-LLaVA. Specifically, on five text-rich multi-image benchmarks, Leopard-LLaVA-Pro outperforms the baseline by an average of +8.79 points.
>
> [1] What If We Recaption Billions of Web Images with LLaMA-3?
>
> [2] Conceptual Captions: A Cleaned, Hypernymed, Image Alt-text Dataset For Automatic Image Captioning
>
> [3] Donut 🍩 : OCR-Free Document Understanding Transformer

---

### Decision · Action_Editor_oaNv · 2025-05-26

**Recommendation:** Accept as is

**Comment:**

In the initial reviews, the reviewers were concerned about missing experiments with LLaMA-3.1-8B, novelty clarification, dataset comparison, and comparison with Qwen2-VL's naïve resolution approach. The provided author rebuttal effectively assuages the reviewer concerns. In the end, all three reviewers are leaning towards accepting the submission. The Action Editor has carefully checked the reviews and author rebuttal, and agrees with the reviewer recommendations.

**Audience:**

The proposed and curated dataset will be of interest to the VLM community to improve the ability of multi-image text-rich visual understanding.

**Claims And Evidence:**

The submission makes the following main contributions:

1. A large-scale instruction-tuning dataset on text-rich multi-image scenarios.
2. An adaptive high-resolution multi-image encoding module.

In the experimental results, the proposed model, Leopard, demonstrates superior performances on text-rich multi-image tasks while maintaining competitive performance on single-image benchmarks.

As a result, the claims are properly supported by the experiments, along with the promise to open-source full dataset and code.